# Photogenerated outer electric field induced electrophoresis of organic nanocrystals for effective solid-solid photocatalysis

Yan Guo [1,2], Bowen Zhu[3], Chuyang Y. Tang [2] ✉, Qixin Zhou [1] ✉ & Yongfa Zhu [1] ✉

Rapid mass transfer in solid-solid reactions is crucial for catalysis. Although phoretic nanoparticles offer potential for increased collision efficiency between solids, their implementation is hindered by limited interaction ranges. Here, we present a self-driven long-range electrophoresis of organic nanocrystals facilitated by a rationally designed photogenerated outer electric field (OEF) on their surface. Employing perylene-3,4,9,10-tetracarboxylic dianhydride (PTCDA) molecular nanocrystals as a model, we demonstrate that a directional OEF with an intensity of 13.6-0.4 kV m$^{-1}$ across a range of 25–200 μm. This OEF-driven targeted electrophoresis of PTCDA nanocrystals onto the microplastic surface enhances the activity for subsequent decomposition of microplastics (196.8 mg h$^{-1}$) into $CO_2$ by solid-solid catalysis. As supported by operando characterizations and theoretical calculations, the OEF surrounds PTCDA nanocrystals initially, directing from the electron-rich (0 1 1) to the hole-rich (11$\bar{2}$) surface. Upon surface charge modulation, the direction of OEF changes toward the solid substrate. The OEF-driven electrophoretic effect in organic nanocrystals with anisotropic charge enrichment characteristics indicates potential advancements in realizing effective solid-solid photocatalysis.

Solid-solid catalysis involves reactions between a solid catalyst and solid reactant at the interface, where collision probability and interfacial contact significantly influence reaction efficiency and selectivity. Examples of solid-solid catalysis include plastic recycling/degradation[1–3], sterilization[4,5], and precious metal recovery[6–8]. In dispersion reactions involving solid particle catalysis, mass transfer constraints, particularly with sizable or highly viscous reactant molecules, result in inadequate contact between reactants and catalysts. Particle catalysts can be directed to solid substrate surfaces via magnetic fields or chemical methods[9,10], illustrating solid-solid interaction distances dictated by short-range phoretic (typically < 10 μm) or van der Waals forces (typically < 1 nm)[11–13]. However, the apparent improvement of solid-solid collision chances by these strategies is still insufficient due to the limited interaction range. Moreover, light-driven movement of nanoparticles focuses on bimetallic catalytic nanowires, microtubular microrockets, and Janus microspheres depends on ion motion but is restricted by fuel requirements, high-energy ultraviolet light irradiation (most studies have focused on $TiO_2$) and the complex design[14–19]. In crystalline photocatalysts, the internal electric field forms anisotropic carrier engagement[20–22], while the outer electric field (OEF), which would affect the migration of catalyst nanocrystals in solution, remains largely unexplored. Inspired by above, we expect to design rationally a photogenerated OEF aim to create long-range electrophoresis force for photocatalyst nanoparticle towards solid substrates thereby enhancing the solid-solid collision probability in the reaction[23]. The organic molecular crystals provide an ideal platform for studying

[1]Department of Chemistry, Tsinghua University, Beijing 100084, China. [2]Department of Civil Engineering, The University of Hong Kong, Hong Kong 999077, China. [3]School of Environmental and Energy Engineering, Beijing University of Civil Engineering and Architecture, 100032 Beijing, China. ✉e-mail: tangc@hku.hk; zqx20@mails.tsinghua.edu.cn; zhuyf@tsinghua.edu.cn

the photogenerated OEF due to the slow-relaxation charge behavior[24–27] and the molecular arrangement depended anisotropic charge transport of organic semiconductors[22,28–31].

In this study, we employ perylene-3,4,9,10-tetracarboxylic acid dianhydride (PTCDA) molecular crystals as a model and combine multiscale simulations with crystal manipulations to uncover the generation rules of photogenerated OEF. Utilizing ultrafast spectroscopy and first-principles calculations, we demonstrate that surface charge regulation enhances the OEF directed from PTCDA molecular nanoparticles to microplastic substrates. Space-time resolved in situ fluorescence and Raman microscopy observed this OEF promotes long-range electrophoresis of PTCDA molecular nanoparticles to microplastic, leading to subsequent microplastic degradation by oxidizing species. Our findings elucidate principles governing crystal surface charge-induced photogenerated OEF and effects on nanoparticle motion, providing novel insights for molecular nanocrystal photocatalyst design and application for solid-solid reaction.

## Results

### Anisotropic charge enrichment on PTCDA nanocrystals

Refinement of the PTCDA crystal model was conducted using powder X-ray diffraction (PXRD) analysis (Supplementary Fig. 1a & Table 1). Cryo-TEM demonstrates the formation of molecular nanocrystals through slippage π-stacking at an angle of 65° and a width of 7.4 Å (Fig. 1a), which corresponds to the (0 1 1) facet of PTCDA nanocrystal. Scanning transmission electron microscopy (STEM) images display the organized streak distribution of π-π stacking at a 3.2 Å distance on the (0 2 0) facet (Fig. 1b). The crystal cell model also presents that the (0 1 1) facet exposes more oxygen, while (11$\bar{2}$) facet involves perylene ring's conjugation plane (Supplementary Fig. 1b–f). Density functional theory (DFT) was employed to show the lowest unoccupied crystal orbitals (LUCO) distribution, unveiling anisotropic charge-enriched characteristics associated with crystal facets. Isosurfaces display

varying LUCO orbital overlap extents on distinct facets, with the largest overlap observed for the (0 1 1) facet. Greater overlap suggests enhanced intermolecular interactions, potentially causing unoccupied energy level splitting and LUCO expansion, thus facilitating electron transition from the highest occupied crystal orbital (HOCO). Conversely, the LUCO of (0 2 0) facet exhibits minimal overlap and the (11$\bar{2}$) displays no overlap under identical conditions (Supplementary Fig. 2), indicating reduced electron receptivity. Electron potential calculations additionally corroborate this through the lower energy barriers of electron migration (Supplementary Fig. 3). Moreover, density of states (DOS) calculations reveals band gaps of 0.78 eV, 1.50 eV, and 1.63 eV for the (0 1 1), (0 2 0), and (11$\bar{2}$) facets, respectively. The narrow bandgap further proves that electrons are preferentially enriched on the (0 1 1) facet. In contrast to the (0 1 1) facet, the (0 2 0) and (11$\bar{2}$) crystal facet display pronounced discrete hole distributions that contribute to the long-term stabilization of holes, signifying a predilection for hole enrichment (Supplementary Fig. 4). These results supporting that the (0 1 1) and (11$\bar{2}$) facets possess the highest propensity for photogenerated electron and hole accumulation on PTCDA nanoparticle surfaces, respectively.

Molecular dynamics predicted crystal morphology aligns with SEM images, guiding the distinguish of crystal facet in atomic force microscopy (AFM) images (Fig. 1c & Supplementary Figs. 5, 6). To visualize the types of photogenerated charges accumulated on the nanocrystal surfaces, we analyzed this correlation by kelvin probe force microscopy to assess surface potential changes under irradiation (Supplementary Note 2)[32–35]. Surface photovoltage microscopy (SPVM) revealed that the surface potential distribution corresponded to crystal facet[36], with (0 1 1) and (01$\bar{1}$) facets accumulating approximately −2 mV photogenerated electron signals, while conjugate (11$\bar{2}$) facets displayed approximately +1.5 mV photogenerated hole signals (Fig. 1d). This electron-rich characteristic exhibited by the (0 1 1) surface was further confirmed by photo-deposition experiments (Pt (IV) is

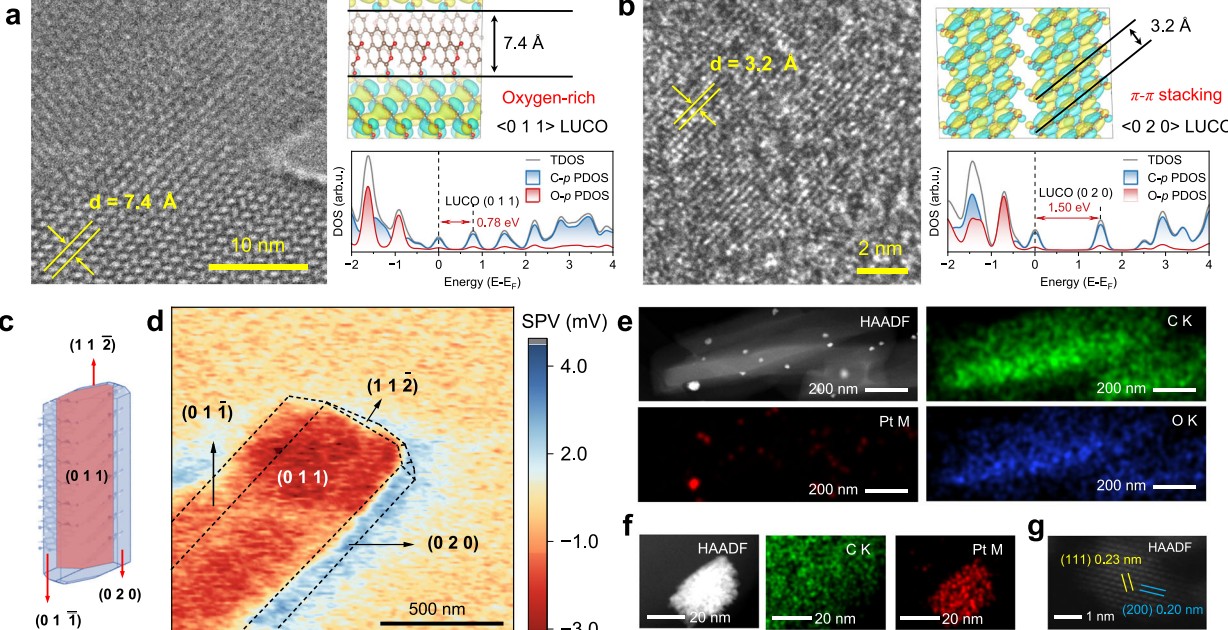

**Fig. 1 | Anisotropic charge enrichment on PTCDA nanocrystals. a** Cryo-transmission electron microscope (Cryo-TEM) image. Right upper: DFT calculations of the LUCO for the (0 1 1) facets. The isosurface of the molecular orbital is set at a value of 0.0075. Right lower: the density of states (DOS) for the (0 1 1) facets. **b** Scanning transmission electron microscopy (STEM) image. Right upper: DFT calculations of the LUCO for the (0 2 0) facets. The isosurface is 0.0075. Right lower: the DOS for the (0 2 0) facets. **c** The molecular packing and crystal

morphology were predicted by molecular dynamics. **d** Surface photovoltage microscope by kelvin probe force microscope. **e** High angle annular dark field–scanning transmission electron microscopy (HAADF-STEM) of PTCDA nanocrystal with photo-deposited Pt nanoparticles. EDS mapping corresponding to Pt, C, and O elements. **f** Another view of HAADF-STEM observations and EDS mapping corresponding to Pt, C elements. **g** Lattice stripes of Pt nanoparticles.

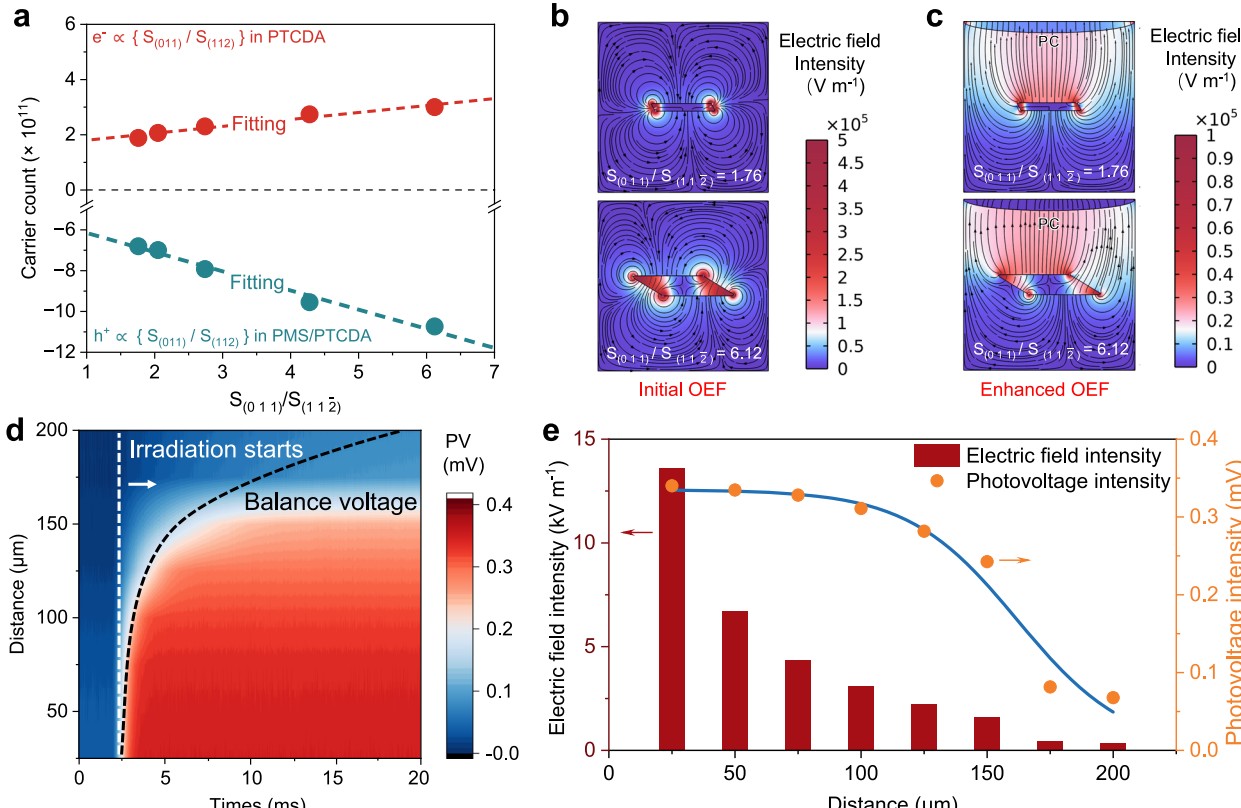

**Fig. 2 | Photogenerated outer electric field (OEF) and its regulation. a** Variation of the photogenerated charge from transient surface photocurrent integral with $(S_{(011)}/S_{(11\bar{2})})$ before and after the addition of electron acceptor PMS. $(S_{(011)}/S_{(11\bar{2})})$ refers to the ratio of the PTCDA (0 1 1) facet area to the $(11\bar{2})$. Data in (**a**) were obtained from integration of the data in Supplementary Fig. 11. Finite element analysis (FEA) of photogenerated OEF on the surface of PTCDA nanocrystal: **b** boundary is 0 mV, (**c**) the upper boundary is set at −1.3 mV. Nanocrystal facet ratio and charge quantities are from Supplementary Figs. 10, 11, respectively. A parallelogram containing (0 1 1) and $(11\bar{2})$ facets was constructed as a PTCDA nanocrystal, the four corners were rounded with a radius of 20 nm, and a square box with a side length of 3.0 μm was used as a simulation domain. In a typical simulation, 33770 domain cells and 908 boundary cells are used. Experimentally tested (**d**) variation of photogenerated potential with distance between PTCDA nanoparticles and PC microplastic (the white dashed line is the irradiation start time and the black dashed line is the equilibrium voltage time, PV photovoltage), (**e**) balance photogenerated voltage and the corresponding calculated OEF strength with distance from (**d**).

reduced to Pt by photogenerated electron). The Pt deposition site is explored utilizing high-angle annular darkfield STEM (HAADF-STEM) atomic imaging. As depicted in Fig. 1e, Pt nanoparticles appeared on the plane and at long edges of PTCDA, while scarcely observed on the shortened edges. The characterization of the elemental distribution by corresponding energy dispersive spectrometer (EDS) further proves this. Since exposure of a significant oxygen concentration to the (0 1 1) facet, the Pt-O bond serves as an indicator for Pt deposition sites, which was confirmed through X-ray photoelectron spectroscopy (XPS) analysis (Supplementary Fig. 7). Alternative HAADF-STEM images perspective (Fig. 1f & Supplementary Fig. 8) also displayed this distribution characteristic of Pt. The lattice fringes of 0.23 nm and 0.20 nm correspond to the (1 1 1) and (2 0 0) facets of the Pt nanoparticles, respectively (Fig. 1g). These investigations reveal that the variability in molecular stacking engenders a propensity for electrons and holes to accumulate at the (0 1 1) and $(11\bar{2})$ facets of PTCDA molecular crystals, respectively. Consequently, the surface of PTCDA nanoparticles exhibits anisotropic electron and hole enrichment characteristics.

### Regulation of photogenerated outer electric field (OEF)

To explore the OEF stemming from photogenerated charge distribution, we first manipulated crystal facets proportion to modulate surface charge density. Employing a top-down liquid phase exfoliation approach, we controlled the proportion of (0 1 1) and $(11\bar{2})$ facets on the surface of PTCDA (Supplementary Fig. 9). SEM and AFM elucidated the exposed facet ratios, revealing an increase in the (0 1 1) to $(11\bar{2})$ surface ratio $(S_{(011)}/S_{(11\bar{2})})$ from 1.76 to 2.05, 2.74, 4.28 and 6.12 (Supplementary Fig. 10). Quantification of photogenerated charge was achieved by integrating transient photocurrent measurements (Supplementary Fig. 11 & Supplementary Table 2). A significant increase in photogenerated electron numbers was observed with higher (0 1 1) facet proportions, displaying an approximate linear relationship. Electron numbers on nanoparticle surfaces rose from $1.88 \times 10^{11}$ to $3.00 \times 10^{11}$ at a rate of $k = 2.5 \times 10^{10}$ (red in Fig. 2a). Multiscale simulations revealed surface-enriched charges distribution (Supplementary Fig. 12). Considering PTCDA's exciton dissociation energy of 38.5 meV (Supplementary Fig. 13–14, the binding energy greater than $kT$ at room temperature), the introduction of an electron acceptor like peroxymonosulfate (PMS), which can capture photogenerated electrons on the surface, leads to a corresponding release of holes. Upon PMS addition, transient surface photocurrent integrals displayed a hole accumulation rate of $k = 9.3 \times 10^{10}$, 3.7 times the electron increase rate associated with the (0 1 1) facet ratio. A total of $10.7 \times 10^{11}$ accumulated holes were observed, exceeding the maximum photogenerated electron number by a factor of 3.57 (green in Fig. 2a), indicating a higher (0 1 1) facet ratio correlates with an increased number of induced corresponding holes. These findings provide strategies for modulating electron and hole quantities on PTCDA nanocrystals' surfaces. The electron-hole accumulation on organic nanocrystal surfaces arises from the inherent ability of organic molecular nanocrystals to circumvent bipolar charge trapping by oxygen and water molecules[26],

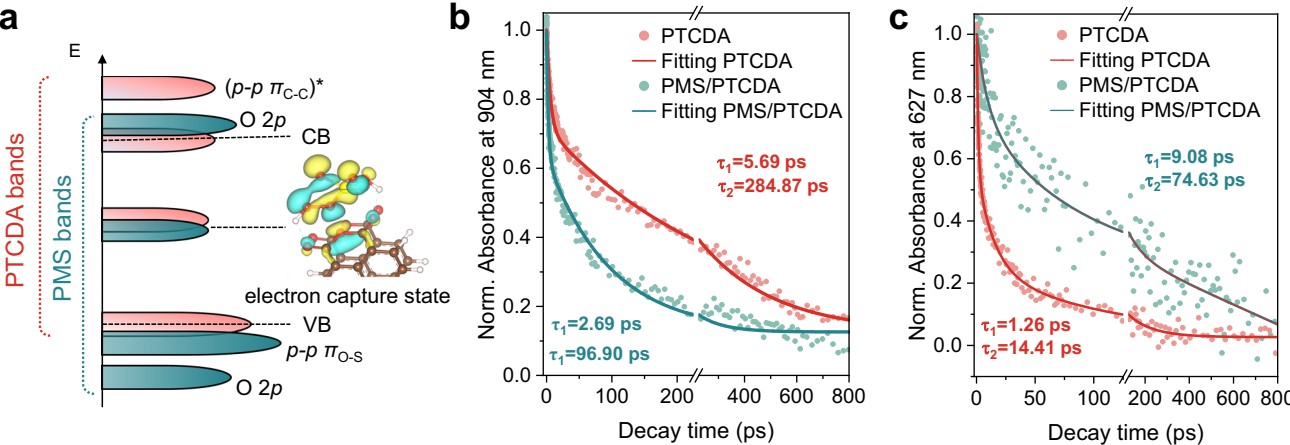

**Fig. 3 | Charge transfer between electron acceptor PMS and PTCDA nanocrystal. a** Scheme of Partial density of states (PDOS) of PMS adsorbed on (0 1 1) facet of PTCDA that Simplified from Supplementary Fig. 21. Attenuation kinetics of femtosecond transient absorption spectra (fs-TAS) under different conditions at (**b**) 904 nm and (**c**) 627 nm.

which emphasizes the universality of charge enrichment-induced OEF within these materials.

The OEF distribution was calculated using finite element analysis (FEA) on (0 1 1) and (11$\bar{2}$) cross-section (Supplementary Fig. 16). As illustrated in Fig. 2b, the initial photogenerated OEF forms a circular configuration around the PTCDA nanocrystal, extending from (11$\bar{2}$) to (0 1 1). And the circular OEF strength along the neutral axis increases with the (0 1 1) facet ratio (Supplementary Fig. 17a). For FEA of hole-occupied surfaces, the upper boundary was modeled at −1.3 mV to emulate a negatively charged solid surface such as polycarbonate (PC) microplastic. As anticipated, the OEF originates from the nanocrystal surface towards the PC, deviating from the initial circular pattern (Fig. 2c). The intensity of this OEF exhibits a correlation with the (0 1 1) facet ratio, which can be ascribed to the augmentation of hole production, resulting from the photogenerated charge separation process facilitated by PMS (Supplementary Fig. 17b). FEA clarifies photogenerated OEF direction and trends, while photovoltage experiments between the PC and nanoparticles allow determining OEF strength (Supplementary Fig. 18).

As depicted in Fig. 2d, the photogenerated voltage of PTCDA nanoparticles achieves equilibrium in milliseconds under 530 nm LED light irradiation, with the equilibrium voltage diminishing as the distance between PTCDA nanoparticles and PC microplastic increases. The strength of the local OEF is estimated by the uniform electric field $\mathbf{E} = U/D$, yielding a strength of 13.6, 6.7, 4.4, 3.1, 2.3, 1.6, 0.5, and 0.4 kV m$^{-1}$ for distances of 25, 50, 75, 100, 125, 150, 175, and 200 μm, respectively (Fig. 2e). Upon linear fitting ($R^2$ = 0.998), the OEF intensity at a 75 μm distance displayed an approximately linear augmentation with PMS concentration up to <2.3 mM, followed by saturation as PMS increased, corroborating the modulation of surface hole-occupied states by PMS (Supplementary Fig. 19). Conversely, without hole-occupied states on PTCDA nanoparticle surfaces, the circular presence interferes with the intensity and distribution of photogenerated OEF (Supplementary Fig. 20).

The comprehensive investigation of charge transfer between electron acceptor PMS and PTCDA nanoparticles entailed verifying the PTCDA-PMS interaction, specifically determining the partial density of states (PDOS) of PMS adsorbed on the (0 1 1) facet by theoretical calculations. The hybridization energy levels of PMS interacting with PTCDA spanned from −1 to 2 eV, displaying a notable hybridization at 1 eV within the forbidden band (Fig. 3a). This result indicates preferential electron acquisition by PMS from HOCO rather than LUCO, which created thermodynamic conditions for PMS extract electron. The orbital diagrams of the corresponding

energy levels shown in the inset also exhibit electron distribution on the PMS. Conversely, there is non-priority acceptance of electronic for (0 2 0) and (11$\bar{2}$) facets (Supplementary Fig. 21). This outcome demonstrates that PMS effectively traps electrons, eliminating exciton confinement of holes and subsequently liberating holes on the PTCDA nanocrystal surface.

Photogenerated electron transfer kinetics between PMS and nanocrystal were investigated using femtosecond transient absorption spectroscopy (fs-TAS) (Supplementary Fig. 22). Near-infrared probing typically detects intraband transitions near the conduction band (CB), rendering the 650-1100 nm probe informative for free/shallow trapped electron dynamics[37]. Consequently, we examined photogenerated electron dynamics at 904 nm. Considering that the valence band (VB) to CB transition of PTCDA molecular crystals corresponds to 1.97 eV (~630 nm) (Supplementary Fig. 23)[38] and the addition of hole quenchers accelerates the ~630 nm kinetics process (Supplementary Fig. 24), it is evident that decay monitoring at 627 nm offers a direct understanding of hole occupancy in the VB. The decay processes of electrons and holes accord with the second-order exponential, and $\tau_1$ and $\tau_2$ are obtained using a global fit (Supplementary Tables 3, 4). $\tau_1$ represents efficient bulk $e^-$ and $h^+$ recombination, while $\tau_2$ denotes carries captured by shallowly trapped states[39]. PMS introduction decreased photogenerated electron $\tau_2$ from 284.87 ps to 96.90 ps (Fig. 3b), indicating effective electrons captures by PMS. Conversely, holes $\tau_1$ extension increased from 1.26 ps to 9.08 ps (Fig. 3e), suggesting PMS inhibits hole-electron coulomb complexation ($\tau_1$). In addition, the plausibility of electron capture by PMS has been verified (Supplementary Figs. 25, 26)[40–42].

## Long-range electrophoresis motion of PTCDA nanocrystal

Microplastics (MPs, particle scize <5 mm) are classified as significant emerging micro-pollutants due to their small size and resistance to biodegradation[43]. Using PC microplastics as a model (Fig. 4a), electrophoretic phenomena of PTCDA nanocrystal were studied. Line scanning confocal reflectance microscopy (LSCRM) probed the dynamic migration behavior of PTCDA nanocrystals around microplastics under different illumination conditions with fine spatio-temporal details, where a focused laser beam is rapidly scanned across a given sample plane and the reflected light is used to build the image (Supplementary Fig. 27). Despite PTCDA dimensions exceeding diffraction limits, the image of the fluorescent signal (red signals based on intrinsic photoluminescence) with high brightness and high stability indicates the position of PTCDA that can be traced over time. Without

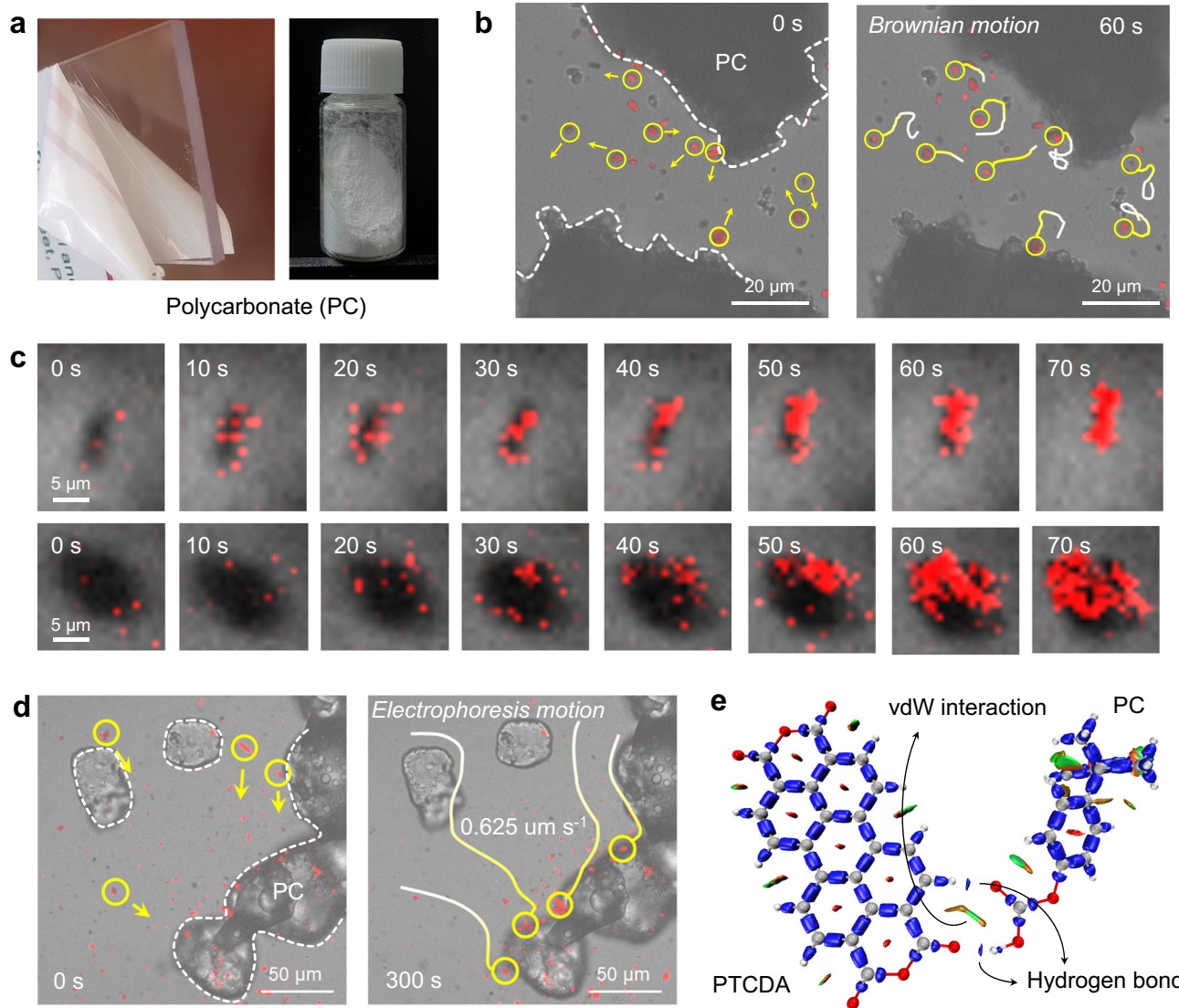

**Fig. 4 | Long-range electrophoresis motion of PTCDA nanocrystal. a** Optical photographs of polycarbonate (PC) sheet and microplastic powders. Confocal fluorescence microscopy observation of PTCDA nanoparticles electrophoresis onto PC microplastics: (**b**) without additional irradiation, (**c**) PTCDA nanoparticles from other focusing layers under irradiation, (**d**) motion trajectory in the focusing layer under irradiation. Gradient lines from white to yellow indicate the time sequence of particle migration. **e** Interaction region indicator (IRI) function analysis of PTCDA and PC molecules.

additional irradiation, PTCDA nanocrystals statistically exhibited random Brownian motion (Fig. 4b). Interestingly, electrophoretically driven PTCDA nanocrystals migrated from other height layers to the focusing surface upon hole-occupied surface induction via electron acceptor addition under light (Fig. 4c). A significant increase in PTCDA nanoparticles aggregated on the plastic surface was observed within 70 s, with untraceable trajectories in the focusing layer due to the origin from other height layers. The collective migration-enrichment behavior of such PTCDA nanocrystals represents the effect of rationally designed photogenerated OEF on bulk particle electrophoresis.

Building on our prior demonstration of tracking directional migration from PTCDA nanocrystals to microplastics under realistic photocatalytic conditions using LSCRM, we further employed this technique to spatially examine the migration rate of nanocrystal trajectories. Figure 4d displays the migration trajectory of PTCDA nanocrystals on the same focusing layer. Under irradiation, nanoparticles swiftly migrated to the PC microplastic surface, amassing in substantial quantities within 300 s. Trajectory lines revealed a peak rate of 5.4 μm s$^{-1}$ and an average electrophoretic mobility of approximately 0.625 μm s$^{-1}$ (Supplementary Fig. 28). These observations indicate that

enhanced OEF through surface charge regulation drives PTCDA nanocrystal long-range electrophoresis, generally augmenting collision probability and reactivity between solid photocatalysts and solid reactive substrates. Concurrently, the stable adsorption following migration is attributed to hydrogen bond contact at the PTCDA-microplastic interface. Interaction region indicator (IRI, Supplementary Note 1) analysis revealed two prominent donor-acceptor hydrogen bonds between PTCDA and PC (Fig. 4e), serving as adsorption sites for electrophoretically deposited PTCDA nanoparticles on the microplastic surface and establishing an efficient charge transfer interface[44].

**Promoted solid-solid photocatalytic reaction**

We employed the particle size of ~0.5 μm PC microplastics (in water as a larger aggregate secondary structure of tens of μm) decomposition process by PTCDA nanocrystal to examine the application of electrophoresis for solid-solid catalysis. As shown in Fig. 5a, PMS/PTCDA photocatalysis achieved 49.2% and 92.3% solid PC removal at 3 and 24 h, respectively. The degradation performance of PC during the first 3 h was 196.8 mg$_{PC}$ h$^{-1}$ in eight reactors. During the 3 h of

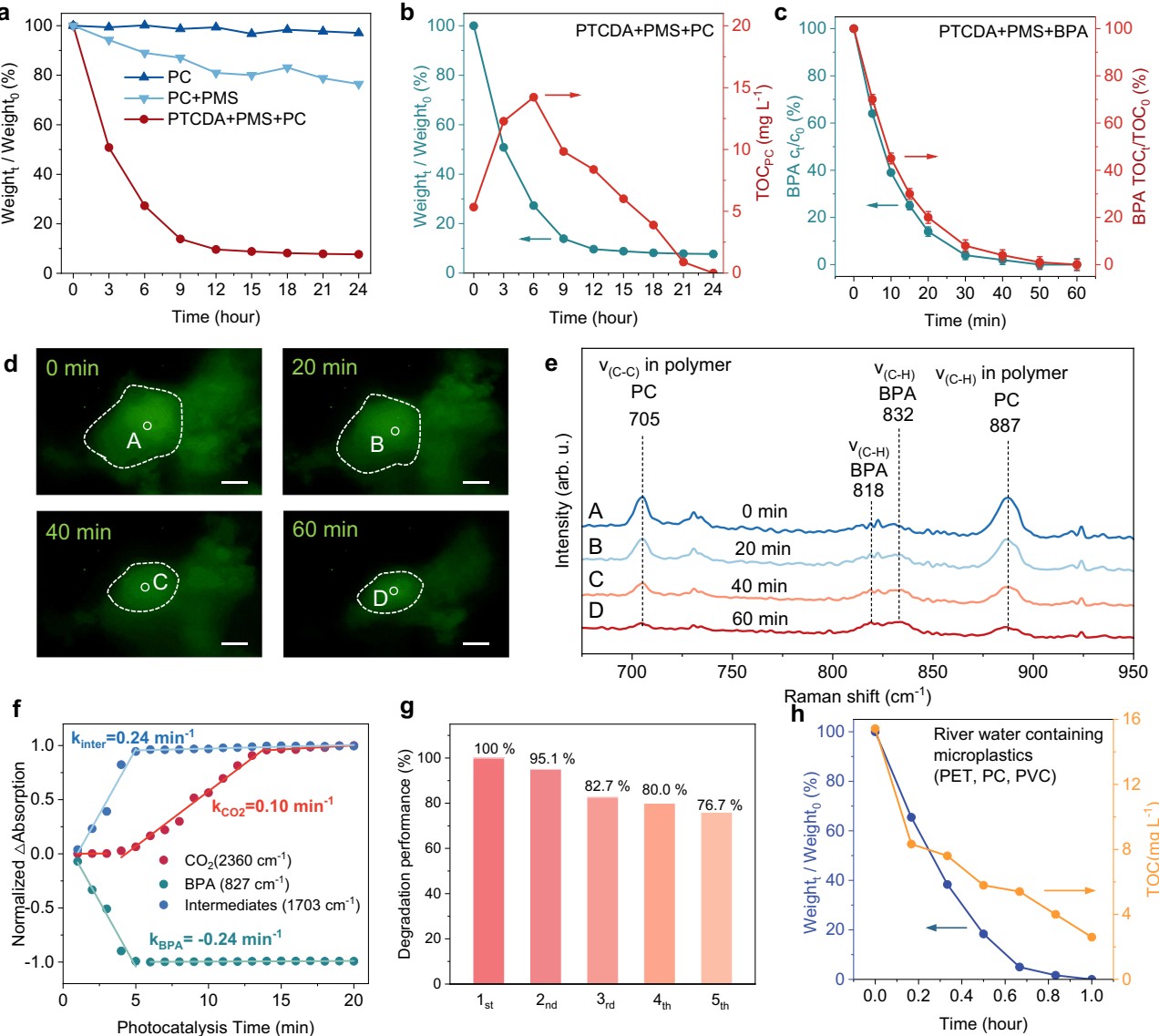

**Fig. 5 | Promoted solid-solid photocatalytic reaction. a** The weight change curve of PC microplastics. The weight$_0$ and weight$_t$ represent the pre-reaction and post-reaction of PC weight, respectively. **b** PC weight and total organic carbon (TOC$_{PC}$) concentration in supernatant. **c** Concentration of BPA and TOC curves in PTCDA photocatalytic oxidation BPA reaction. The error bars represent the standard deviation after three individual experiments. **d** Confocal Raman microscopy in situ observation of PC microplastic degradation and (**e**) in situ Raman spectra of the labeled regions in (**d**). **e** Normalized absorption intensities of 2360, 827, 1703 cm$^{-1}$ vary with photocatalysis time. The data points in (**f**) were extracted from Supplementary Fig. 34. **g** Assessing photocatalyst stability through cyclic degradation experiments: 5 Cycles at 24 h each, measuring efficacy retention. **h** Purification of river water containing PET, PC, and PVC microplastics.

photodegradation in the PMS/PTCDA system, the PC microplastics showed irregular and small fragments of nanoscale (Supplementary Fig. 29). Moreover, the diminishing intensity of PC characteristic peaks, as observed via attenuated total reflection Fourier transform infrared (ATR-FTIR) analysis[45], indicates the progressive decomposition of PC microplastics (Supplementary Fig. 30). Total organic carbon (TOC$_{PC}$) content of the supernatant (Supplementary Note 3) was assessed to investigate the thoroughness of PC microplastics being oxidatively decomposed[46]. The TOC value reached a maximum of 14.22 mg L$^{-1}$ at 6 h and diminished to zero by 24 h (Fig. 5b), implying that PC microplastics initially underwent decomposition into smaller organic molecules, followed by 100% completely oxidation into CO$_2$ and H$_2$O. We performed several additional control experiments to compare with the photoactivity of the PMS/PTCDA system. Firstly, PC dispersion in water underwent aging tests under simulated light irradiation, resulting in 2.9% mass loss after 24 h, attributed to light irradiation and mechanical stirring[47,48]. Secondly, PMS addition

accelerated PC hydrolysis likely due to saline environment corrosion, causing 23.6% mass loss after 24 h[49].

DFT calculations of bond energy and electrostatic potential substantiate the ready formation of depolymerization products in PC plastics by BPA monomers (Supplementary Fig. 31), in accordance with hydrolysis products identified in previous studies[50–52]. Remarkably, complete BPA oxidation to CO$_2$ and H$_2$O occurred within 60 min under irradiation by PMS/PTCDA (Fig. 5c). We propose that directional electrophoresis of PTCDA and subsequent efficient oxidation of BPA underlies the depolymerization of PC microplastics. We employed in-situ confocal Raman microscopy, providing simultaneous microscopic images and real-time Raman spectra under irradiation, to visualize the degradation of solid PC microplastics (Fig. 5d). Among them, the outline of microplastics in the field of view gradually became smaller in the observation 60 min, indicating rapid microplastic solids removal. At the same time, the Raman characteristic peaks of PC corresponding to 705 cm$^{-1}$ and 887 cm$^{-1}$ gradually decreased during degradation,

while peaks of BPA analogs corresponding at 818–832 cm$^{-1}$ increased (Fig. 5e), signifying PC microplastic degradation into smaller units accompanying PC segment depolymerization.

Diminished PMS concentrations resulted in a decline in PC decomposition (84.1%), while continuous PMS addition achieved 100% decomposition (Supplementary Fig. 32a). The observed decline in PMS concentration suggests its involvement in PC degradation (Supplementary Fig. 32b), including both the modulation of the PTCDA surface's charge and the generation of reactive oxygen radicals. These reactive oxygen radicals for the degradation of PC were identified as the PMS products $OH^{\cdot}$ and $SO_4^{\cdot-}$, as well as the released photogenerated holes (Supplementary Fig. 33). Concentration variations were determined by ATR-FTIR using characteristic peaks for $CO_2$, BPA, and intermediates, identified at 2360, 827, and 1703 cm$^{-1}$, respectively (Supplementary Fig. 34)[53]. A synchronous decrease in BPA concentration and an increase in carbonyl products during the first 5 min with rate constant of 0.24 min$^{-1}$ suggest that the direct products of BPA oxidation involve carbonyl (Fig. 5f & Supplementary Tables 5, 6). The expeditious emergence of the $CO_2$ peak at 0.1 min$^{-1}$ within a 4-min interval signifies the system's robust and thorough oxidizing potential for BPA.

Products in gases and liquids during PC degradation were identified, and an energy rationalization of the transition state using DFT was also performed to propose a pathway for PC decomposition (Supplementary Figs. 35–37). The findings reveal that PTCDA photocatalysis rapidly and completely oxidation the depolymerization monomer BPA of PC plastics, contributing to the complete removal of PC plastics from water. Moreover, cyclic stability experiments assessed the stability of the PTCDA photocatalyst. As depicted in Fig. 5g, after five cycles (cumulative 120 h), the degradation of PC microplastics persisted at 76.7%. The stability of the PTCDA heterogeneous photocatalyst is further evidenced by its oxidation capabilities for various water pollutants in a continuous flow reactor, highlighting its potential for scalable applications (Supplementary Note 4).

The broad adoptability of this long-range electrophoretic behavior has been confirmed over microplastics with compositions in actual river water, including polyvinyl chloride (PVC), polyethylene terephthalate (PET) and PC, size (~0.5–20 μm, Supplementary Fig. 38), concentration (0.3 mg L$^{-1}$), and surface charge conditions (negative zeta potential). Owing to electrophoresis motion and oxidizing species, complete decomposition of microplastics is achieved with low plastic concentrations (Fig. 5h). Given the high initial TOC value of 15.4 mg L$^{-1}$ in the river water (Supplementary Table 7), the 83.0% reduction in TOC highlights the remarkable oxidizing capacity of the PMS/PTCDA system.

In conclusion, this study reveals a prevalent photogenerated OEF on crystals established by distinct electron and hole distributions, enabling the electrophoresis of nanocrystals. Surface charge regulation disrupts the circular OEF generating an enhanced OEF, observed 13.6 kV m$^{-1}$ (axial component at 25 μm) for the PTCDA photocatalyst, that drives nanocrystal long-range electrophoretic mobility. Directing nanoparticles to accumulate around microplastics in water facilitated unparalleled efficiency in solid-solid photocatalytic reactions. These findings pioneer the comprehension of photogenerated OEF on photocatalyst crystals, highlighting the paramount importance of surface charge modulation rules on OEF.

## Methods
### Experimental
The reagents employed in the synthesis of photocatalysts possess a purity of 99.9%. Reagents of analytical purity are utilized in the performance evaluation. Details of the Kelvin probe force microscope (KPFM), total organic carbon (TOC) measurements and principles, and 0.5 m$^2$ scale reactor are described in the electronic supplementary note 2, 3, and 4.

## Characterization
Cryo-TEM (G4i, cryo-TEM), high-resolution transmission electron microscopy (HR-TEM, JEOL JEM-2100F), field emission scanning electron microscopy (FE-SEM, Hitachi SU-8010), and atomic force microscopy (AFM, SPM-9700) (with high-resolution probe and Kelvin Probe Force Microscopy (KPFM) mode) were used to study the morphology of the samples. HAADF-STEM tested by spherical aberration corrected transmission electron microscopy (FEI, Themis Z). The catalytic products in the liquid phase were analyzed by high-performance liquid chromatography-time-of-flight mass spectrometry (LCMS-TOF, Shimadzu). Total organic carbon concentrations were obtained using a German Jena (multi N/C2100 TOC). Bisphenol A concentrations were analyzed by SHIMADZU (LC-20AT). A Bruker D8 Advance X-ray diffractometer obtained PXRD of the samples at 45 kV and 200 mA using Cu Kα1 ($\lambda = 1.5418$ Å). Fourier transform infrared spectra (FTIR) with attenuated total reflection (ATR) were recorded on a Bruker V70 spectrometer. The Raman signal was analyzed using a laser confocal Raman spectrometer (Horiba HR-800). In situ Raman spectra for the photocatalytic process were acquired under 400 nm monochromatic intense light excitation, circumventing incident light interference while simultaneously achieving material photoexcitation. UV-Vis diffuse reflectance spectroscopy (DRS) spectra were obtained by a Hitachi U-3010 spectrophotometer (reference BaSO4). Laser scanning confocal microscope (Olympus, FV300RS), configured with 50 mW at 405 nm, 20 mW at 488 nm, 20 mW at 561 nm, and 40 mW at 640 nm, was used to test the trajectories of molecular nanoparticles. X-ray photoelectron spectra (XPS) were obtained using a Thermo Escalab 250XI instrument with a 1.6 A current. Charge correction was performed by setting the C1s peak of contaminated carbon to 284.8 eV. Unless otherwise specified, measurements were made using PTCDA with a (0 1 1) crystal plane to (11$\bar{2}$) crystal facet ratio of 6.12 in all fs-TAS, FTIR, DRS, XPS, and photocatalytic degradation tests.

**Visible-mid-infrared femtosecond transient absorption spectroscopy (fs-TAS).** Fs-TAS is employed to examine photocatalysts' excited state dynamics and ultrafast spectral processes. A TOPAS Prime (Spectra Physics) optical parametric amplifier generates an excitation light pulse, pumped by a Ti-sapphire femtosecond laser (35 fs, 800 nm, 1 kHz). The output pulse is divided into two beams. The first undergoes second harmonic generation via a nonlinear optical crystal, yielding a 400 nm laser pulse. The second, attenuated beam is focused on a 4 mm-thick CaF4 plate, producing white light as the probe. An electronic time delay line controls the pulse delay. Detected light variations are captured by a fiber-optic spectrometer (AVaspec-ULS2048CL-EVO, Avantes). UV-Vis-NIR TAS measurements of PTCDA aqueous dispersions, with and without PMS, are conducted using 1 mm pathlength quartz cuvettes.

**Quantification of photogenerated charges on the PTCDA surface.** The photogenerated charge density on the PTCDA surface was determined through temporal integration of the transient photocurrent. Transient photocurrent measurements were performed in a fixed-capacitor arrangement. Photocurrent transients were excited with pulses from a tunable white LED (1400 W m$^{-2}$). The transients were detected by a CIMPS-fit transient system (Zahner) with a resolution time of 50 ns. The device configuration in all tests was ITO-sample-ITO, where the sample layer thickness was about 1 μm. In particular, the PTCDA-PMS sample was pre-centrifuged after the solution phase adsorbed PMS, and other test conditions were consistent with PTCDA.

**Photogenerated outer electric field strength measurement.** A PTCDA nanoparticle film was prepared on ITO glass via drop-coating to create one electrode, while the second electrode was fabricated by hot-melting PC microplastics onto ITO glass at 230 °C. The inter-electrode distance was regulated using multiple layers of 25 μm thick

3 M tape. The effective working electrode area was 2.25 cm². For distance-influenced OEF intensity tests, the employed electrolyte involved 5 mM PMS. For the OEF intensity test for the effect of PMS concentration, the electrode distance was controlled to be 75 μm. Without PMS, the electrolyte was replaced with 5 mM Na₂SO₄. The photogenerated voltage was evaluated under 530 nm LED lamp illumination utilizing the aforementioned electrochemical workstation. See Supplementary Fig. 17 for a schematic diagram of measurement device.

## DFT calculations

The density of electronic state calculations was performed using DFT, as implemented in the CASTEP code. DFT total energy calculations were performed using the spin-polarized Perdew-Burke-Ernzerhof exchange-correlation functional[54]. The latest Heyd-Scuseria-Ernzerhof (HSE06) hybrid functional[55] described the electronic exchange-correlation energy. An energy cut-off of 500 eV was used for identifying the accuracy of the calculations. A convergence tolerance of $1.0 \times 10^{-5}$ eV atom⁻¹ is used for the electronic structure iteration. The selected calculation parameters were all tested to ensure that energy convergence was less than 1 meV per atom. The geometry optimizations, and electrostatic potential analysis of PC polymers (B3LYP-D3 6-311 + G* level) were performed using the Gaussian 09 program, where the bond energy was obtained by Laplace bond level extrapolation. M06-2X functional gives refined energies for organic systems.

**Prediction of crystal morphology.** Energy minimization was performed using molecular dynamics method, a cut-off distance of 1.55 nm, and a force field of consistent valence forcefield. The Growth Morphology algorithm was used to predict the morphology of PTCDA molecular crystals under vacuum.

## Preparation of PTCDA

Different (0 1 1) crystalline and (11$\bar{2}$) crystalline ratios of PTCDA are prepared by a top-down method of liquid phase exfoliation. Specifically, 50 mg of purchased commercial PTCDA was taken and dispersed in 10 mL of sulfuric acid at concentrations of 1.2, 2.3, 4.6, 9.2, and 18.4 mol L⁻¹, respectively. The dispersion was sonicated for 1 h then 100 mL of deionized water was added. The pH was adjusted to 7 by washing with extraction and finally dried under vacuum at 50 °C for 8 h.

## PC microplastic degradation

An aqueous PC dispersion of 3 g L⁻¹ was obtained by mechanical mixing. As needed, 2 g L⁻¹ of PTCDA powder and 10 mM of PMS were added to the PC solution. A PCX-50C multichannel photocatalytic reaction system was employed to simultaneously irradiate eight reactors (50 mL) with an intensity of ~100 mW cm⁻² (Supplementary Fig. 39). Periodically, 20 mL of the dispersion was centrifuged and the supernatant was analyzed for total organic carbon (TOC_PC). The precipitate was dried at 70 °C and weighed to determine the residual PC microplastics mass. Degradation efficiency was estimated by monitoring changes in plastic weight.

## Data availability

The data that supports the findings of the study are included in the main text and supplementary information files. Source Data file has been deposited in Figshare under accession code https://doi.org/10.6084/m9.figshare.24866022[56].

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

## Acknowledgements

We thank F. Wei (THU, Center of Pharmaceutical Technology) for her help with mass spectrometry analysis. Y. Weng, Z. Wang, and Y. Xu (all IOP) for their support on the transient absorption spectroscopy tests. We are grateful to R. Zong, S. Yue, and C. Ma (THU, Analysis Center) for their help with the electron microscopic analysis. This work was supported by National Science Foundation of China (22136002), National Key Research and Development Project of China (2020YFA0710304).

## Author contributions

Y.G., Y.Z. and C.Y.T. designed this research project. Y.G. and C.Y.T. co-wrote the paper. Q.Z. and B.Z. carried out the sample synthesis, characterization, and DFT calculations. Y.G. and Q.Z. designed and executed photocatalytic and electrochemical experiments. Q.Z. performed multi-scale modeling, analysis and calculations. All the authors discussed the results.

## Competing interests

The authors declare no competing interests.
