## [Peer Review File · Nature Communications]

Photogenerated outer electric field induced electrophoresis of organic nanocrystals for effective solid-solid photocatalysisREVIEWER COMMENTS

Reviewer #1 (Remarks to the Author):

This article presents a novel finding that the anisotropic charge enrichment of organic nanocrystalline PTCDA leads to electrophoretic migration, which effectively enhances photocatalytic solid-solid reactions. This investigation into the phenomenon and characteristics of anisotropic charge enrichment in organic semiconductors bridges the knowledge gap in charge migration research within the realm of organic photocatalysts.

The pioneering exploration of external electric fields in photocatalysts described in this paper showcases the potential application of these electric fields in solid-solid catalytic reactions. From scientific phenomena to practical applications, the results discussed in this paper are both innovative and captivating. Furthermore, the authors employ an outdoor flat plate reactor to demonstrate the remarkable performance and stability of the photocatalyst. Some in situ characterizations provide sufficient evidence to prove the points in this manuscript.

Despite these advantages, there are still some minor issues in the text that require revision, especially clarification of experimental details. In summary, I recommend that this article be accepted for publication in Nature Communications with minor revisions. My specific comments are summarized below.

1. The authors should explain more clearly the test conditions in the spectral tests (especially the transient absorption spectra of Fig. 3b and c). Is it a solution phase test or a direct test in the solid phase? Is PMS in dissolved form or is it adsorbed on the surface of nanocrystals in solid form?
2. The single nanoparticle tracing technology demonstrated by the author is very novel in the field of photocatalysis, but some details have not been well clarified. It is foreseeable that the sample preparation process and testing process have a great impact on the test results. Relevant discussion should be added in the revision.
3. I noticed that in Supplementary Figure 13, Temperature-dependent photoluminescence detects the exciton binding energy. The rationality of this detection technology requires more demonstration or additional relevant literature support. And, will organic nanocrystal PTCDA undergo irreversible phase transition at such a low temperature?
4. Figure 5d and e refer to the in situ confocal fiber Raman technique and the Raman spectroscopy used in the supporting information. I can't seem to find the configuration of the relevant instruments and the setting conditions for the light cycle. Also, the light spectrum and irradiation intensity used in different experiments should be discussed.
5. CO and CH₄ were detected in Supplementary Figure 33. The origin of this gas phase product is not well discussed. Is it from CO₂ reduction after mineralization of microplastics? Or is it from structural fragments of microplastics and further hydrogenation?
6. In Figure 5a the author shows the optimal plastic degradation conditions, but at the same time the real-time PMS concentration should be given as a verification of the mechanism. In addition to the role of modulating crystal surface electrical properties emphasized by the author, as a well-known potential oxidative species, to what extent does excess PMS contribute to the plastic degradation reaction?

Reviewer #2 (Remarks to the Author):

I read the manuscript by Guo et al. with interest, which describes a fascinating new discovery related to the external electric field of organic photocatalysts. The authors demonstrate that a regulated outer electric field can drive the electrophoresis of the photocatalyst, and they observe that this property significantly improves solid-solid reactions. The authors propose a novel scientific understanding of the photogenerated external electric field, providing a new paradigm for future research on photocatalytic solid-solid reactions.

This study provides a deeper understanding of the relationship between the crystal structure and charge properties of organic photocatalysts, filling a research gap in the field of organic photocatalysts. More important, the authors have applied this significant finding to solid-solid reactions and achieved an enhancement, offering new prospects for organic crystals. A variety of different structural and photophysical techniques, as well as theoretical calculations, are employed to gain a better understanding of the system. The work presents high-quality data and contributes new scientific findings concerning organic crystals, migration behavior, and charge dynamics of organic photocatalysts. Furthermore, I believe that many of these findings have the potential to be transferred to other systems, thus attracting a wide audience. Hence, I would like to recommend it to be accepted after a minor revision, as some issues require further clarification and discussion. The detailed comments and discussions are listed below:

1. To provide a more comprehensive understanding of the outer electric field's role, it would be beneficial to additionally analyze its effect on electron-hole separation within the photocatalyst bulk phase and on charge migration. Although the impact of the built-in electric field on the migration of each carrier is mentioned in the introduction, a more detailed analysis would enhance the manuscript.
2. Figure 5b shows the total organic carbon content in the liquid, peaking at around 6 minutes and then gradually decreasing. This timescale aligns with the BPA mineralization rate to CO₂ shown in Figure 5f. The related discussion could be expanded in this section.
3. It is worth noting that the flattened sensory map in Supplementary Figure 22 makes it challenging to discern the excitation of electrons, and there is a need for more finely tuned isosurfaces.
4. How does the concentration of PMS influence the external electric field? To further validate the physicochemical processes proposed by the authors, it is essential to demonstrate the presence of a saturating critical PMS concentration, beyond which the external electric field is no longer enhanced.
5. The horizontal labeling in Figure 5h should be consistent with the other graphs.
6. Figure 5h illustrates the removal of PET and PVC microplastics and mentions their sizes in the discussion. It would be helpful to exhibit electron microscope images for PET and PVC in the supporting information.
7. The manuscript refers to a calculation of 196.8 mg PC/h, but the details of this calculation are not provided. Please include this information in the methods section.
8. In the Conclusion, the unit of 13.6 kV should be corrected to kV/m.
9. The methodology for the confocal fluorescence microscopy test in Figure 4 should be provided in the Methods section. Specifically, how is the PC immobilized on the substrate?
10. The numbers in Supporting Form 1 should include units.
11. The "Cumulative induced holes of PMS/PTCDA" in Supporting Form 2 does not appear to be necessary. Consider removing or revising this part.

Response to the reviewers' comments

Manuscript Title: Anisotropic Charge Enrichment Induced Organic Nanocrystals Photocatalyst Electrophoretic Motion for Promoting Solid-Solid Catalysis

Manuscript Number: NCOMMS-23-41933

Author: Yan Guo, Bowen Zhu, Chuyang Y. Tang, Qixin Zhou and Yongfa Zhu

[The reviewer comments are shown in *italic*; responses are in blue; all revisions in the main text and supplementary information are highlighted in red]

Reviewer #1

General comment: *This article presents a novel finding that the anisotropic charge enrichment of organic nanocrystalline PTCDA leads to electrophoretic migration, which effectively enhances photocatalytic solid-solid reactions. This investigation into the phenomenon and characteristics of anisotropic charge enrichment in organic semiconductors bridges the knowledge gap in charge migration research within the realm of organic photocatalysts.*

The pioneering exploration of external electric fields in photocatalysts described in this paper showcases the potential application of these electric fields in solid-solid catalytic reactions. From scientific phenomena to practical applications, the results discussed in this paper are both innovative and captivating. Furthermore, the authors employ an outdoor flat plate reactor to demonstrate the remarkable performance and stability of the photocatalyst. Some in situ characterizations provide sufficient evidence to prove the points in this manuscript.

Despite these advantages, there are still some minor issues in the text that require revision, especially clarification of experimental details. In summary, I recommend that this article be accepted for publication in Nature Communications with minor revisions. My specific comments are summarized below.

Response: We thank reviewer 1 for his positive comments on the manuscript and are grateful to reviewer 1 for giving us the opportunity to revise and clarify experimental details.

Comment 1: *The authors should explain more clearly the test conditions in the spectral tests (especially the*

transient absorption spectra of Fig. 3b and c). Is it a solution phase test or a direct test in the solid phase? Is PMS in dissolved form or is it adsorbed on the surface of nanocrystals in solid form?

Response: Thanks to your reminder, we have added the method description of TAS in the Method section of the manuscript (page 19 of the manuscript):

Visible-mid-infrared femtosecond transient absorption spectroscopy (fs-TAS)

Fs-TAS is employed to examine photocatalysts' excited state dynamics and ultrafast spectral processes. A TOPAS Prime (Spectra Physics) optical parametric amplifier generates an excitation light pulse, pumped by a Ti-sapphire femtosecond laser (35 fs, 800 nm, 1 kHz). The output pulse is divided into two beams. The first undergoes second harmonic generation via a nonlinear optical crystal, yielding a 400 nm laser pulse. The second, attenuated beam is focused on a 4 mm-thick CaF₄ plate, producing white light as the probe. An electronic time delay line controls the pulse delay. Detected light variations are captured by a fiber-optic spectrometer (AVaspec-ULS2048CL-EVO, Avantes). UV-Vis-NIR TAS measurements of PTCDA aqueous dispersions, with and without PMS, are conducted using 1 mm pathlength quartz cuvettes.

Comment 2: *The single nanoparticle tracing technology demonstrated by the author is very novel in the field of photocatalysis, but some details have not been well clarified. It is foreseeable that the sample preparation process and testing process have a great impact on the test results. Relevant discussion should be added in the revision.*

Response: Thanks to the very careful scrutiny of the reviewers, we have added the test procedure in the Supplementary Figure 27 section describing the line scanning confocal reflectance microscopy test (page 31 of the Supplementary Information):

PTCDA exhibits significant fluorescence, enabling observation of its particle behavior through laser scanning confocal microscopy. PC microplastics were heat-melted and affixed to 20 mm diameter glass-bottomed confocal petri dishes. PTCDA was dispersed in aqueous solutions, either with or without PMS. The petri dish was positioned under a laser confocal microscope, and PTCDA dispersion was added dropwise in situ to monitor migration behavior. A 561 nm low light intensity served as the fluorescence excitation source for PTCDA particles, while a combination of 405 nm, 488 nm, 561 nm, and 640 nm high-intensity light sources functioned as the irradiation source.

Comment 3: *I noticed that in Supplementary Figure 13, Temperature-dependent photoluminescence detects the exciton binding energy. The rationality of this detection technology requires more demonstration or additional relevant literature support. And, will organic nanocrystal PTCDA undergo irreversible phase*

transition at such a low temperature?

Response: Thanks to the reviewer's check, we have added references to Temperature-dependent fluorescence testing of exciton binding energy:

The exciton binding energy (E_b) is calculated by Equation 1^{2,3}:

2. Shi, Y.; Li, J.; Mao, C.; Liu, S.; Wang, X.; Liu, X.; Zhao, S.; Liu, X.; Huang, Y.; Zhang, L., Van Der Waals gap-rich BiOCl atomic layers realizing efficient, pure-water CO₂-to-CO photocatalysis. *Nat. Commun.* **2021**, *12* (1), 5923.

3. Li, C.; Liu, J.; Li, H.; Wu, K.; Wang, J.; Yang, Q., Covalent organic frameworks with high quantum efficiency in sacrificial photocatalytic hydrogen evolution. *Nat. Commun.* **2022**, *13* (1), 2357.

In addition, we investigate the crystalline phase of PTCDA nanocrystals at low temperatures. Low-temperature XRD demonstrated <1% change indicating that the PTCDA crystalline phase is stable at low temperatures. Therefore, it is reasonable to test the exciton binding energy (Supplementary Figure 13) by low-temperature fluorescence:

Page 17 of Supplementary information:

Supplementary Figure 14. Low temperature XRD of PTCDA. (a) Low index diffraction peak changes with temperature, and (b) the corresponding interplanar spacing changes.

The change amplitude of the low-index XRD diffraction peak is within 1%, indicating that the crystal structure has not changed significantly.

Comment 4: Figure 5d and e refer to the in situ confocal fiber Raman technique and the Raman spectroscopy used in the supporting information. I can't seem to find the configuration of the relevant instruments and the setting conditions for the light cycle. Also, the light spectrum and irradiation intensity used in different experiments should be discussed.

Response: We appreciate the reviewer's reminder and have supplemented the Raman testing conditions and the employed light source spectrum:

Page 18-19 of manuscript:

The Raman signal was analyzed using a laser confocal Raman spectrometer (Horiba HR-800). In situ Raman spectra for the photocatalytic process were acquired under 400 nm monochromatic intense light excitation, circumventing incident light interference while simultaneously achieving material photoexcitation.

Supplementary Figure 39. Spectrograms used in this work

Comment 5: CO and CH₄ were detected in Supplementary Figure 33. The origin of this gas phase product is not well discussed. Is it from CO₂ reduction after mineralization of microplastics? Or is it from structural fragments of microplastics and further hydrogenation?

Response: Although the research on CO₂ photoreduction has made great progress in recent years, it is difficult to drive the reduction of CO₂ to other products due to the lack of suitable catalytic sites on the surface of PTCDA. Although thermodynamically seemingly satisfactory, the potentially extremely high overpotential makes this process inert. To this end, we additionally tested the CO₂ reduction activity of PTCDA in a humid CO₂ atmosphere.

PTCDA is almost inert in CO₂ reduction under humid CO₂ atmosphere. (Test conditions: 10 mg PTCDA, 99.9999% CO₂, humidity 100%. Irradiation intensity: ~ 1Sun.) Taking into account the enhanced stability of

PTCDA, the above discussion shows that the gas phase products generated originate from plastics rather than CO₂ reduction.

Page 42 in Supporting Information:

Supplementary Figure 36. Gaseous products of PTCDA (a) photocatalytic degradation of PC microplastics and (b) photocatalytic reduction of CO₂ tested by Shimadzu gas chromatography (GC-2014)

PTCDA is almost inert in CO₂ reduction under humid CO₂ atmosphere. (Test conditions: 10 mg PTCDA, 99.9999% CO₂, humidity 100%. Irradiation intensity: ~ 1Sun.) Taking into account the enhanced stability of PTCDA, the tested CO and CH₄ shows that the gas phase products generated originate from plastics rather than CO₂ reduction.

Comment 6: In Figure 5a the author shows the optimal plastic degradation conditions, but at the same time the real-time PMS concentration should be given as a verification of the mechanism. In addition to the role of modulating crystal surface electrical properties emphasized by the author, as a well-known potential oxidative species, to what extent does excess PMS contribute to the plastic degradation reaction?

Response: Thanks to the reviewers' comments, in this revision, we have added information on the variation of PMS concentration during degradation and discussed the effect of PMS on the PC degradation process:

Page 14 of manuscript:

...PMS addition accelerated PC hydrolysis likely due to saline environment corrosion, causing 23.6% mass loss after 24 hours⁴⁶...

Page 16 of manuscript:

The observed decline in PMS concentration suggests its involvement in PC degradation (Supplementary Fig. 32b), including both the modulation of the PTCDA surface's charge and the generation of reactive oxygen radicals. These reactive oxygen radicals for the degradation of PC were identified as the PMS products OH·

and $\text{SO}_4^{\cdot-}$, as well as the released photogenerated holes (Supplementary Fig. 33).

Page 37 of Supporting Information:

Supplementary Figure 32. The change of (a)PC microplastics mass and (b) PMS concentration during photocatalytic degradation of microplastics by PMS/PTCDA.

The residual PMS concentration was ascertained via colorimetric analysis. A color-developing solution was prepared, consisting of 0.5 g NaHCO_3 , 10 g KI, and 100 mL deionized water. In standard tests, 1 mL of diluted filtrate was combined with 1 mL of the color-developing solution at specified intervals. After 15 minutes, PMS concentration was determined using a UV-Vis spectrophotometer at 400 nm.

Supplementary Figure 33. Identification of active species during degradation. (a) Electron paramagnetic resonance (EPR) signal of PTCDA aqueous dispersion with or without PMS under in-situ irradiation ($\lambda \geq 420$ nm). (b) Effect of active species scavengers on the degradation rate of BPA.

Reviewer #2

General comment: I read the manuscript by Guo et al. with interest, which describes a fascinating new discovery related to the external electric field of organic photocatalysts. The authors demonstrate that a regulated outer electric field can drive the electrophoresis of the photocatalyst, and they observe that this property significantly improves solid-solid reactions. The authors propose a novel scientific understanding of the photogenerated external electric field, providing a new paradigm for future research on photocatalytic solid-solid reactions.

This study provides a deeper understanding of the relationship between the crystal structure and charge properties of organic photocatalysts, filling a research gap in the field of organic photocatalysts. More important, the authors have applied this significant finding to solid-solid reactions and achieved an enhancement, offering new prospects for organic crystals. A variety of different structural and photophysical techniques, as well as theoretical calculations, are employed to gain a better understanding of the system. The work presents high-quality data and contributes new scientific findings concerning organic crystals, migration behavior, and charge dynamics of organic photocatalysts. Furthermore, I believe that many of these findings have the potential to be transferred to other systems, thus attracting a wide audience. Hence, I would like to recommend it to be accepted after a minor revision, as some issues require further clarification and discussion. The detailed comments and discussions are listed below:

Response: We express our gratitude to the reviewers for their positive recognition of the innovation in our work and appreciate the opportunity provided to revise and further clarify and supplement the discussion.

Comment 1: To provide a more comprehensive understanding of the outer electric field's role, it would be beneficial to additionally analyze its effect on electron-hole separation within the photocatalyst bulk phase and on charge migration. Although the impact of the built-in electric field on the migration of each carrier is mentioned in the introduction, a more detailed analysis would enhance the manuscript.

Response: We appreciate the reviewer's suggestions. In this work, PMS acts as an electron acceptor, capturing photogenerated electrons on the surface of the photocatalyst PTCDA, thereby enabling holes to occupy the surface and ultimately establishing an outer electric field (OEF) between the photocatalyst and the substrate. During the OEF establishment process, the recombination of electron-hole pairs is suppressed. Therefore, the separation of electron-hole pairs is a necessary and sufficient condition for the formation of the OEF. We apologize for not clearly describing this physical mechanism in the original version, and to make it clearer, we have added the following explanation in the revised manuscript:

Page 9 of manuscript:

The intensity of this OEF exhibits a correlation with the (0 1 1) facet ratio, which can be ascribed to the augmentation of hole production, resulting from the photogenerated charge separation process facilitated by PMS (Supplementary Fig. 17b).

In addition, in order to make the discussion of hole dynamics in the text more comprehensive, we have supplemented this revised version with a further attribution of the hole signal from the fs-TAS:

Page 10-11 in manuscript:

Considering that the valence band (VB) to CB transition of PTCDA molecular crystals corresponds to 1.97 eV (~630 nm) (Supplementary Fig. 23)³⁸ and the addition of hole quenchers accelerates the ~630 nm kinetics process (Supplementary Fig. 24), it is evident that decay monitoring at 627 nm offers a direct understanding of hole occupancy in the VB.

Supplementary Figure 24. The hole quencher ascorbic acid identifies the hole signals of TAS.

The addition of the hole quencher ascorbic acid significantly reduced the delay of the ~630 nm signal, indicating that it is reasonable to analyze the hole kinetics with ~630 nm.

Comment 2: Figure 5b shows the total organic carbon content in the liquid, peaking at around 6 minutes and then gradually decreasing. This timescale aligns with the BPA mineralization rate to CO₂ shown in Figure 5f. The related discussion could be expanded in this section.

Response: We appreciate the reviewers' feedback. As noted, Figure 5b displays a peak TOC time of 6 min for the PC degradation process, while Figure 5f demonstrates BPA mineralization to CO₂ via FTIR monitoring at 5 min. Despite temporal proximity, the kinetics of these processes are incommensurable due to distinct test

conditions and degraded substances. Specifically, the mineralization process monitored by FT-IR features test conditions outlined in Supplementary Figure 34.d. The obtained kinetics of BPA mineralization are solely applicable to this film's test conditions, with anticipated differences in solution-dispersed state kinetics.

Comment 3: It is worth noting that the flattened sensory map in Supplementary Figure 22 makes it challenging to discern the excitation of electrons, and there is a need for more finely tuned isosurfaces.

Response: Thanks to the reviewer for pointing this out, we have re-rendered Supplementary Figure 25 (the original version of Supplementary Figure 22) to make it more three-dimensional and intuitive.

Supplementary Figure 25. The isosurface of the differential charge density by TDDFT (Yellow is charge enriched and blue has reduced electron density).

Comment 4: How does the concentration of PMS influence the external electric field? To further validate the physicochemical processes proposed by the authors, it is essential to demonstrate the presence of a saturating critical PMS concentration, beyond which the external electric field is no longer enhanced.

Response: Thanks to the reviewer's suggestion, we tested the effect of PMS concentration on OEF intensity and added relevant descriptions in the manuscript and supporting information:

Page 9 of Manuscript:

Upon linear fitting ($R^2=0.998$), the OEF intensity at a 75 μm distance displayed an approximately linear augmentation with PMS concentration up to <2.3 mM, followed by saturation as PMS increased, corroborating the modulation of surface hole-occupied states by PMS (Supplementary Fig. 19).

Page 23 of Supplementary information:

Supplementary Figure 19. The OEF intensity tested for the effect of PMS concentration. (a) Photovoltage intensity as affected by different PMS concentrations. (b) Relationship between PMS concentration and electric field strength. The distance between the PTCDA and PC microplastic electrodes was fixed at 75 μm . The effective working electrode area was 2.25 cm^2 .

Comment 5: The horizontal labeling in Figure 5h should be consistent with the other graphs.

Response: Thanks to the reviewer's careful examination, we have changed "hours" to "min" to keep the horizontal coordinates of Figure 5h consistent with the other figures.

Comment 6: Figure 5h illustrates the removal of PET and PVC microplastics and mentions their sizes in the discussion. It would be helpful to exhibit electron microscope images for PET and PVC in the supporting information.

Response: Thanks to the reviewers for comments. According to your suggestion, we have supplemented the testing of PET and PVC morphology and added in Supplementary:

Supplementary Figure 38. SEM images PET and PC microplastics used in this work.

Comment 7: The manuscript refers to a calculation of 196.8 mg PC/h, but the details of this calculation are not provided. Please include this information in the methods section.

Response: Thanks to reviewer's comments. We add the description about the details of this degradation performance:

Page 13 of Manuscript:

The degradation performance of PC during the first 3 hours was 196.8 mg_{PC}/h in eight reactors.

Page 21 of Manuscript:

An aqueous PC dispersion of 3 g L⁻¹ was obtained by mechanical mixing. As needed, 2 g L⁻¹ of PTCDA powder and 10 mM of PMS were added to the PC solution. A PCX-50C multichannel photocatalytic reaction system was employed to simultaneously irradiate eight reactors (50 mL) with an intensity of ~100 mW cm⁻².

Comment 8: In the Conclusion, the unit of 13.6 kV should be corrected to kV/m.

Response: Thanks to the reviewer for careful scrutiny. We have corrected this to kV/m.

Comment 9: The methodology for the confocal fluorescence microscopy test in Figure 4 should be provided in the Methods section. Specifically, how is the PC immobilized on the substrate?

Response: Thanks to the reviewer's comment. According to your suggestion, we have add the details of tests:

Page 31 of Supplementary information:

PTCDA exhibits significant fluorescence, enabling observation of its particle behavior through 1 line scanning confocal reflectance microscopy (LSCRM). PC microplastics were heat-melted and affixed to 20 mm diameter glass-bottomed confocal petri dishes. PTCDA was dispersed in aqueous solutions, either with or without PMS. The petri dish was positioned under a laser confocal microscope, and PTCDA dispersion was added dropwise in situ to monitor migration behavior. A 561 nm low light intensity served as the fluorescence excitation source for PTCDA particles, while a combination of 405 nm, 488 nm, 561 nm, and 640 nm high-intensity light sources functioned as the irradiation source.

Comment 10: The numbers in Supporting Form 1 should include units.

Response: Thanks to the reviewer's carefully examination, we add the units in the Supplementary Table 1:

Supplementary Table 1. Simulated cell parameters of PTCDA molecular crystals

Lattice type	a (Å)	b (Å)	c (Å)	α (°)	β (°)	γ (°)
Monoclinic	4.500	14.700	12.000	90.000	91.480	90.000

Comment 11: The "Cumulative induced holes of PMS/PTCDA" in Supporting Form 2 does not appear to be necessary. Consider removing or revising this part.

Response: Thanks to the reviewer's comment, we have deleted the "Cumulative induced holes of PMS/PTCDA" in Supplementary Table 2 according to the suggestions.

Supplementary Table 2. Fitting parameters for the relationship between the electronic quantity and $\{S_{011}\} / \{S_{11\bar{2}}\}$ in Figure 2a.

	Equation	k	b	R2
Electrons in PTCDA	$y=kx+b$	2.5×10^{10}	1.6×10^{11}	0.94
Holes in PMS/PTCDA		9.3×10^{10}	5.2×10^{11}	0.98

REVIEWERS' COMMENTS

Reviewer #1 (Remarks to the Author):

All my concerns are well addressed. The manuscript is now acceptable in Nature Comm.

Reviewer #2 (Remarks to the Author):

The authors have addressed all my concerns and I recommend publication of the manuscript in its current form.